# Molecular-Targeted Therapy for Tumor-Agnostic Mutations in Acute Myeloid Leukemia

**DOI:** 10.3390/biomedicines10123008

**Published:** 2022-11-22

**Authors:** Hironori Arai, Yosuke Minami, SungGi Chi, Yoshikazu Utsu, Shinichi Masuda, Nobuyuki Aotsuka

**Affiliations:** 1Department of Hematology and Oncology, Japanese Red Cross Narita Hospital, Iidacho 286-0041, Japan; 2Department of Hematology, National Cancer Center Hospital East, Kashiwa 277-8577, Japan

**Keywords:** acute myeloid leukemia, tumor agnostic, solid tumor, genomic profiling, hereditary breast and ovarian cancer, variant, molecular-targeted therapy, BRCA

## Abstract

Comprehensive genomic profiling examinations (CGPs) have recently been developed, and a variety of tumor-agnostic mutations have been detected, leading to the development of new molecular-targetable therapies across solid tumors. In addition, the elucidation of hereditary tumors, such as breast and ovarian cancer, has pioneered a new age marked by the development of new treatments and lifetime management strategies required for patients with potential or presented hereditary cancers. In acute myeloid leukemia (AML), however, few tumor-agnostic or hereditary mutations have been the focus of investigation, with associated molecular-targeted therapies remaining poorly developed. We focused on representative tumor-agnostic mutations such as the *TP53*, *KIT*, *KRAS*, *BRCA1*, *ATM*, *JAK2*, *NTRK3*, *FGFR3* and *EGFR* genes, referring to a CGP study conducted in Japan, and we considered the possibility of developing molecular-targeted therapies for AML with tumor-agnostic mutations. We summarized the frequency, the prognosis, the structure and the function of these mutations as well as the current treatment strategies in solid tumors, revealed the genetical relationships between solid tumors and AML and developed tumor-agnostic molecular-targeted therapies and lifetime management strategies in AML.

## 1. Introduction

In the recent past, the available treatments for patients with acute myeloid leukemia (AML) produced a poor prognosis and consisted only of cytotoxic agents without indication for radical operations or radiotherapies. Recently, a variety of tumor-agnostic mutations have been found, and novel molecular-targeted therapies have been developed that have rapidly improving the prognosis of patients with solid tumors. For example, EGFR (epidermal growth factor receptor) inhibitors were developed about twenty years ago and approved in Japan as molecular-targeted therapies for non-small-cell lung cancer (NSCLC); since then, a variety of EGFR inhibitors have been developed worldwide [1]. Moreover, other kinds of EGFR inhibitors targeting colorectal cancers and head and neck cancers have also been developed [2]. Thus, various drugs targeting specific mutations have been successively developed across different tumor types. Moreover, TRK (tropomyosin receptor kinase) inhibitors such as larotrectinib and entrectinib are symbolic of a new era, in which tumor-agnostic therapeutics take the lead for cancer drug therapy. TRK inhibitors have been recently approved for tumors with *NTRK* (neurotrophic tyrosine receptor kinase 3 gene) fusions, which are rare but are detected in multiple tumor types including salivary gland, thyroid, colon, lung, gastrointestinal stroma, appendix, breast, pancreas, sarcoma, melanoma and cholangiocarcinoma with *NTRK* gene fusions, showing highly positive results with larotrectinib treatment [3]. The development of tumor-agnostic molecular-targeted therapies will undoubtedly accelerate in the future. In addition, immune checkpoint inhibitors (ICIs) have recently been approved for several cancers [4,5,6,7,8,9,10,11]. A proportion of patients with stage IV or relapsed malignant tumors that received ICIs are still alive more than five years later, whereas more than a decade ago, the life expectancy of patients with progressed cancer was generally expected to be approximately one year [12]. Moreover, genomic status such as microsatellite instability–high (MSI–High) and tumor mutation burden (TMB) has also recently become a target of ICIs, indicating that all kinds of cancers can be treated by ICIs [8,13]. However, several clinical trials with ICIs for AML have not yet produced the expected results [14]. The currently known genetic targets in solid tumors, for which specific agents are approved by the Food and Drug Administration (FDA), are summarized in Table 1 [15]. Comprehensive genomic profiling (CGP) is a next-generation sequencing technique where hundreds of cancer-related genes are concurrently examined, and it enables the detection of various tumor-agnostic mutations presumed to influence tumor development. CGP has enabled the rapid accumulation of information on tumor-agnostic gene mutations and a huge number of clinical trials have been conducted, resulting in the development of molecular-targeted therapies for various solid tumors. However, the development of therapies for hematological tumors is lagging. It is known that the types and structures of mutations are somewhat different between solid tumors and hematological tumors. First, the number of mutated genes detected in hematological tumors is much smaller than that of solid tumors [16]. AML is considered to comprise seven representative abnormalities, such as signaling genes, DNA methylation-associated genes, myeloid transcription factor gene fusion or mutations, chromatin-modifying genes, nucleophosmin gene, tumor-suppressor genes, splicesome complex genes, and cohesion complex genes [17]. Most of these genetic abnormalities are difficult to detect by conventional examination, which has made the development of new treatments challenging. However, it is now known that AML also has some mutations that are common with solid tumors and that these tumor-agnostic mutations can be candidates of new molecular-targeted therapies. Further, mutations of hereditary tumors, such as hereditary breast and ovarian cancer (HBOC), have often been incidentally identified, though the relationship between these mutations and AML remains unknown. In this review, we summarized the frequency, the prognosis, the structure and the function of these mutations as well as current treatment strategies in solid tumors, revealed the genetical relationships between solid tumors and AML and developed tumor-agnostic molecular-targeted therapies and lifetime management strategies in AML.

## 2. (HM)-SCREEN-Japan 01

(HM)-SCREEN-Japan 01 is a multicenter genomic observational study [18,19]. Patients with AML relapsed/refractory to initial therapy or those newly diagnosed but intolerable to standard therapy were recruited. The objective of this study was to evaluate the frequency and characteristics of cancer-related genetic abnormalities in AML patients detected by a comprehensive genome profiling assay. A detailed analysis of published data detected nine typical tumor-agnostic mutations, including *TP53* (tumor suppressor protein p53 gene), *KIT* (tyrosine-protein kinase gene), *KRAS* (kirsten rat sarcoma viral oncogene homologue gene), *BRCA1* (breast cancer gene 1), *ATM* (ataxia telangiectasia gene), *JAK2* (janus kinase 2 gene), *NTRK3*, *FGFR3* (fibroblast growth factor receptor 3 gene) and *EGFR* (Figure 1). These results encouraged us to review the frequency, prognosis, structure and function of these mutations and their applications to treatment strategies in AML. We focused on these representative tumor-agnostic mutations but excluded the mutations mainly detectable in AML such as FLT3 and IDH1, because the purpose of this review was to consider if the molecular-targeted drugs for solid tumors can be applicable for AML. 

We also showed a schematic drawing of cell proliferation (Figure 2). JAK2, KIT, EGFR, NTRK3 and FGFR3 proteins are tyrosine kinase receptors. KRAS proteins transduce receptor signaling into cellular processes. These proteins activate the signaling pathways such as PI3K/AKT/mTOR and RAS/RAF/MEK/ERK pathways. BRCA and ATM have an important role in repairing damaged DNA. Molecular-targeted drugs suppress these proteins overexpressed by gene mutations. *TP53* maintains the harmony between cell arrest and growth during genomic stress. Approved drugs for *TP53* mutations actually do not exist even for solid tumors, although some important studies are ongoing.

## 3. TP53

### 3.1. Frequency and Prognosis

*TP53* is known as a guardian of genomes, and its mutations are associated with decreased survival in AML patients. *TP53* mutations have been reported in 8% of all AML cases, 4.3% of de novo AML, 16.7% of secondary AML, 25–45% of post-myeloproliferative neoplasm AML (MPN-AML) and 70% of AML with complex karyotype AML [20,21,22,23,24]. TP53 represents a powerful adverse indicator of poor prognosis independent of complex karyotype, age and other genetic markers. *TP53* mutations have also been reported in 14% of patients with therapy-related myelodysplastic syndrome (MDS) and AML [24]. Overall survival (OS) and AML transformation in MDS were significantly different between the TP53 allelic state. In the multi-hit state, the median OS was 8.7 months, compared with 2.5 years in monoallelic patients [25]. *TP53* mutations are more common in therapy-related MDS (t-MDS) and are associated with a high risk of progression to AML [26]. TP53-mutated t-MDS patients more frequently had multiple hits compared to TP53-mutated de novo patients (84 versus 65%) [27,28]. Patients with acute erythroid leukemia (AEL) frequently had *TP53* gene mutations (25–36%), and biallelic mutations are a feature of a subset of AEL [29]. TP53-mutations are associated with very poor prognosis as well in AEL (median survival of 13 months, with no patients surviving at 5 years) [30,31].

### 3.2. Structure and Function

Normal (wild-type, WT) p53 protein encoded by the *TP53* gene is located on chromosome 17p13.1. The functional loss of p53 is common in aggressive advanced cancers. The *TP53* gene is largely a stress response protein with functions ranging from apoptosis to cell cycle control. Therefore, a mutation in the *TP53* gene amplifies the effects of oncogenes leading to the unregulated growth of tumor cells. These mutations are known to result in a gain of function (GOF), a loss of function (LOF) or a non-mutational dysfunction or inactivation of the p53 protein [31]. p53 inactivation involves multipathways, resulting in decreased p53 levels and subsequent cell growth. This inactivation is a direct result of *TP53* gene mutation in one TP53 allele, which can eventually lead to the loss or partial inactivation of the other WT allele over time or during disease progression, owing to the loss of heterozygosity (LOH). LOH is considered a genetic mutation that further develops tumor progression in the context of somatic and germline mutations [32]. *TP53* mutations in leukemia occur in the context of somatic and germline mutations, the latter being associated with Li–Fraumeni syndrome, which leads to the development of certain solid tumors. However, sporadic *TP53* mutations appear to occur upon exposure to carcinogens (environmental factors) or genotoxic insults. This observation further reveals the robust functionality of p53 in maintaining genomic stability and preventing tumorigenesis [33,34,35].

### 3.3. Potential Treatment Strategies

There are no approved therapies to address *TP53* mutations. In a prospective clinical trial of decitabine (10-day courses) to treat AML or transfusion-dependent MDS, patients with *TP53* mutations achieved a significantly higher clinical response rate (100%, 21/21) than patients with wildtype TP53 (41%, 32/78). Furthermore, patients with TP53-mutated AML or MDS experienced similar overall survival (OS) to those with wildtype TP53 (12.7 vs. 15.4 months, *p* = 0.79) [36]. In contrast, historical data on standard chemotherapy treatment associate *TP53* mutation with poor outcomes [31,37,38]. An independent retrospective study also suggests that *TP53* mutation in MDS may predict complete response (CR) to decitabine, but in this report, *TP53* mutation was associated with a shorter OS, even among patients with CR (14 vs. 39 months, *p* = 0.012) [39]. In addition, *TP53* mutation has been associated with lower CR and CR with incomplete marrow recovery (CRi) rates in AML patients treated with venetoclax monotherapy, venetoclax plus chemotherapy or venetoclax in combination with decitabine as compared to wildtype TP53 patients in these settings [40,41]. A number of novel agents are being investigated in this patient group [42]. Of these, APR-246 has evoked considerable excitement based on its robust clinical efficacy in TP53 mutant MDS/AML patients. APR-246, a methylated PRIMA-1 analog, is a small molecule that selectively induces apoptosis in TP53 mutant cancer cells in combination with azacytidine in in vivo models [42,43]. In the AML TCGA dataset, patients with *TP53* mutations exhibited elevated PD-L1 expression relative to those with unmutated TP53 [32], which may lead to the development of ICIs for TP53-mutated AML in the future.

## 4. KIT

### 4.1. Frequency and Prognosis

*KIT* mutations have been reported in more than 90% of cases of mast cytosis [44], 80–85% of cases of gastrointestinal stromal tumor (GIST) [45], 10–20% of cases of melanoma [46] and 4–17% of cases of AML [47], primarily core-binding factor AML (CBF-AML; ~30%) [48]. Reports on the prognostic value of *KIT* mutations in AML have been mixed [49,50], but some studies have found that patients with *KIT* mutations have a poorer prognosis than those without [49,51,52].

### 4.2. Structure and Function

The KIT gene is located on human chromosome segment 4q11 and consists of 21 exons [52]. The KIT protein is a receptor tyrosine kinase and contributes to signal transduction in hematopoietic stem cells, mast cells and the Cajal cells of the digestive tract [53]. KIT encodes a cell surface tyrosine kinase receptor that activates the phosphoinositide 3-kinase/protein kinase B (PI3K/AKT) and rat sarcoma virus/mitogen-activated protein kinase (RAS/MAPK) signaling pathways [54]. KIT aberrations, including point mutations, translocations, amplification and overexpression, have been associated with various malignancies, and KIT is considered as an oncoprotein. KIT signaling has the roles of both downregulation and upregulation. In many cancers, including GIST and AML, KIT has been reported to be activated in the form of overexpression or mutation. Activating alterations in KIT, or the related gene PDGFRA, are known to underlie an inherited predisposition to GIST development [55,56,57]. *KIT* mutations within the activation loop (A-loop), including the amino acids C809, D816, D820, N822, Y823 and A829, have been reported to confer preclinical and clinical resistance to imatinib and sunitinib in GIST [48,58,59]. KIT D816G has also been reported as an emergent mutation conferring resistance to crizotinib in a patient with ROS1-rearranged NSCLC [60]. In a rare case, a PR to imatinib was observed in a patient with a KIT A-loop mutation [61]. KIT exon 17 mutations, including at D816, showed clinical sensitivity to avapritinib [62].

### 4.3. Potential Treatment Strategies

On the basis of clinical evidence, primarily in GIST, AML and systemic mastocytosis, KIT-activating alterations are associated with sensitivity to KIT tyrosine kinase inhibitors (TKIs) including imatinib, sunitinib, sorafenib, dasatinib, nilotinib, ponatinib, midostaurin and avapritinib [63,64,65]. Imatinib is only approved for GIST and mastocytosis by the FDA at present. Avapritinib selectively inhibits mutated KIT and PDGFRA [62] and has demonstrated clinical activity against KIT exon 17 mutations and PDGFRA exon 18 mutations [66,67,68]. Preclinical [69,70] and limited clinical [71,72] data indicate that KIT A-loop mutations are sensitive to sorafenib, although preclinical evidence of potential resistance has been reported specifically for the D816V mutation [70,73]. Several A-loop alterations at residues D816, D820, N822 and A829 have exhibited ponatinib sensitivity in preclinical evaluations [59,74,75,76]. The use of mTOR inhibitors as an alternative therapeutic strategy has demonstrated some success in KIT-activated melanoma, thereby suggesting that mTOR inhibitors as a monotherapy or in combination with first-line kinase inhibitors may be effective in targeting kinase-inhibitor-resistant tumors [77].

## 5. KRAS

### 5.1. Frequency and Prognosis

*KRAS* mutations have been reported in 4–25% of AML cases, in 2% of AML associated with MDS and at varied frequency (0–11%) in other subgroups of AML [78,79,80]. *KRAS* mutations are associated with distinct cytogenetic sub-groups but are not independent prognostic indicators of AML patient outcome. Studies have reported that *KRAS* mutation had no influence on clinical outcome in pediatric patients with AML, in AML patients under 60 years old, in patients with secondary AML or in patients with AML harboring CBFB-MYH11 fusions [81,82].

### 5.2. Structure and Function

There are three functional RAS genes, *H-RAS*, *K-RAS* and *N-RAS*. They cause the constitutive activation of RAS-proteins-transducing receptor signals for cellular processes such as proliferation, differentiation and apoptosis [80,83,84,85]. They have been identified in many types of cancer. KRAS alterations affecting the amino acids G12, G13, Q22, P34, A59, Q61 and A146, as well as the mutations G10_A11insG, A18D, L19F, D33E, G60_A66dup/E62_A66dup and K117N have been characterized to be activating and oncogenic [85,86,87,88,89,90,91,92]. *GOF* mutations in the KRAS pathway are the most common form of AML in adults, including activating mutations of the upstream receptor tyrosine kinases FLT3 and KIT [17]. Each of these mutations produces altered proteins that directly or indirectly drive RAS GTPase into a constitutively active GTP-bound state and leads to the constitutive activation of MAPK/PI3K pathways [93].

### 5.3. Potential Treatment Strategies

In a phase II trial, sotorasib led to a durable clinical benefit in patients with previously treated KRAS p.G12C-mutated NSCLC and was approved by the FDA [94]. Sotorasib is a small molecule that specifically and irreversibly inhibits KRAS G12C. Sotorasib covalently binds to a pocket of the switch II region that is present only in the inactive GDP-bound conformation, trapping KRAS G12C in the inactive state and inhibiting KRAS oncogenic signaling. Other preclinical evidence suggests that KRAS activation may predict sensitivity to MEK inhibitors, such as trametinib and cobimetinib [85,95,96,97]. Clinical responses to MEK-inhibitor-based therapy regimens have been observed in patients with *KRAS* mutation in certain tumor types [83,84,98]. The reovirus Reolysin targets cells with activated RAS signaling [99] and is being evaluated in clinical trials of patients with various tumor types. Reolysin has demonstrated mixed clinical efficacy, with the highest rate of response reported for patients with head and neck cancer [100,101,102].

## 6. BRCA1

### 6.1. Frequency and Prognosis

*BRCA1* mutations have been observed in <0.5% of acute myeloid leukemia (AML) cases (COSMIC, Dec 2018) [17]. Reduced BRCA1 expression was reported at high frequency in therapy-related AML, largely due to BRCA1 promoter hypermethylation, which was observed in 38% of samples [103]. Rare cases of treatment-related myeloid leukemia have been reported in patients with breast and ovarian cancer after drug therapy, including platinum-based agents. *BRCA1* mutations have also been implicated in the development of leukemia. A family history of breast or ovarian cancer is associated with an increased risk of leukemia in breast cancer patients, and *BRCA1* mutations have been implicated in leukemogenesis [104]. Germline mutation in BRCA1 is associated with breast–ovarian cancer familial susceptibility, also known as hereditary breast and ovarian cancer (HBOC) [105,106]. The lifetime risk of breast and ovarian cancer in *BRCA1* mutation carriers has been estimated to be as high as 87% [107], and elevated risks of other cancer types, including gastric, pancreatic, prostate and colorectal, have also been identified, with a frequency range of 20–60% [108]. The estimated prevalence of deleterious germline *BRCA1* mutations in the general population is between 1:400 and 1:800, with an approximately 10-fold higher prevalence in the Ashkenazi Jewish population [109,110]. In the appropriate clinical context, germline testing of BRCA1 is recommended.

### 6.2. Structure and Function

BRCA1 has an important role in maintaining genomic integrity, especially in rapidly dividing cells. In BRCA-deficient cells, the alternate, error prone, double strand DNA break–repair mechanisms predominate, leading to genomic instability [111]. BRCA1 alterations that disrupt the ring-type zinc finger domain (amino acids 24–65) or BRCT domains (aa1642-1855) are predicted to result in a loss of function [112]. The BRCA1 protein is composed of 24 exons and 1866 amino acids. There are some hotspots where many pathogenic gene variants of solid cancers are located. Breast cancer cluster regions (BCCR) and ovarian cancer cluster regions (OCCR) are the representative hotspots where the majority of hereditary breast and ovarian cancer genes are located (Figure 2) [113].

### 6.3. Potential Treatment Strategies

BRCA1-deficient tumor cells are sensitive to inhibitors of poly (ADP-ribose) polymerase (PARP1) through the mechanism of synthetic lethality. Several PARP inhibitors have now received FDA approval for breast, ovarian and pancreatic cancers and mCRPC. Alterations that inactivate BRCA1 confer sensitivity to PARP inhibitors, such as olaparib, rucaparib or niraparib, and to DNA-damaging drugs, such as cisplatin or carboplatin [114,115,116,117,118]. Clinical response to PARP inhibitors has been reported for solid cancer patients with either germline or somatic *BRCA1* mutations [116,117]. A preclinical study confirmed the low levels of BRCA1 expression in AML primary samples and cell lines and suggested that this low expression may underlie the sensitivity of these samples to the PARP inhibitor olaparib [119]. However, there are few reports that consider if PARP inhibitors can be promising for the treatment of AML with somatic or germline BRCA mutations in clinical settings. Unlike other oncogenes that gain pathogenesis by driver mutations, BRCA as a tumor suppressor gene gains pathogenesis according to the locations and the types of mutation. The discovery of hereditary tumors with improved genomic examinations has also increased the significance of assessment and management for germline mutations. In (HM)-SCREEN-Japan, five cases with somatic or germline BRCA1 variants (Figure 3, shown with the numbers in small orange circles) were detected. Positions of these variants are shown below with red vertical lines on the BRCA1 protein. Four variants were located on BCCR. Three cases (cases 1, 3 and, 5) were considered as germline because of high variant allele frequencies (0.438, 0.495 and 0.489, respectively). Four cases (cases 2, 3, 4 and 5) were considered as pathogenic according to ClinVar (a public archive of reports of the relationships among human variations and phenotypes). Case 1 was also located on BCCR, suggesting its pathogenicity. Thus, all cases were considered as germline or pathogenic, or both. However, the karyotypes of these cases were relatively preserved with a few chromosomal changes and with other variants, such as MLL, FLT3, NPM1, TET2, IDH1, DNMT3 and TP53, which are well-known to influence the development and prognosis of AML. BRCA genes encode a tumor suppressor that, when inactivated, contributes to the pathogenesis of breast and ovarian cancers through genome instability. When BRCA gene variants lose their function as tumor suppressors, they allow the accumulation of DNA damage that ultimately destroys the structure of chromosomes. Thus, these cases were recognized to develop AML by co-mutations rather than *BRCA1* mutations that appeared to be detected incidentally. They concluded that the BRCA1 variants found in this study did not play a major role in AML development. However, the accumulation of findings is needed to evaluate the relationships between *BRCA1* mutation and AML development.

### 6.4. Fanconi Anemia

Fanconi anemia (FA), an inherited chromosomal instability disorder, is caused by at least 22 germline pathogenic gene mutations (*PGVs*) including *BRCA1/2* [121]. Most PGVs are autosomal recessive. The FA genes code for proteins that comprise a complex network for DNA damage repair (FA/BRCA DNA repair pathway) and other cellular processes [122,123]. The main function of the pathway is the removal of DNA inter-strand crosslinks, which interfere with DNA replication and gene transcription [124]. FA is characterized by congenital and developmental abnormalities and a strong cancer predisposition. The classical clinical phenotype of FA presents with microcephaly, thumb and radial ray abnormalities, short stature and café au lait spots. Affected patients with severe phenotypes need complex interventions for multiple congenital malformations and malignant complications early in life. Most FA patients present progressive bone marrow failure in the first and second decades of life, and FA-associated bone marrow failure can transform into AML. Patients with FA have an extremely high risk of developing malignancies at an early age; the most common are AML and squamous cell carcinomas of the head and neck and of the female genitalia. Less than 5% of FA cases are caused by bi-allelic PGVs of BRCA2, and very rare cases are caused by bi-allelic PGVs in BRCA1 [125]. FA with BRCA2 patients are at high risk of solid tumors and refractory leukemia [126]. Bone marrow failure and transformation into AML has not been reported with FA-like clinical features caused by bi-allelic BRCA1 PGVs. FA caused by BRCA1/2 PGVs is strongly associated with distinct spectra of embryonal childhood cancers, AML with BRCA2-PGVs and early epithelial cancers with BRCA1 PGVs.

## 7. ATM

### 7.1. Frequency and Prognosis

In the TCGA dataset, *ATM* mutation or homozygous deletion was not observed in any of the 200 AML cases [17]. In the COSMIC database (Feb 2019), *ATM* mutation has been reported in 7.6% of AML samples. *ATM* mutation has been identified in 1.8% of acute lymphoblastic leukaemia (ALL) cases, 1% of B-cell ALL (B-ALL) cases and <1% (1/416) of T-cell ALL (T-ALL) cases analyzed in the COSMIC database (Feb 2019). One study reported that a single nucleotide polymorphism in ATM (rs3092856) was associated with worse outcomes in AML [127]. *ATM* mutations have been reported in 60% of children with T-ALL but were also found in 52% of controls; however, studies have isolated biologically significant alterations in the coding sequences of ATM that are suggested to lead to the development of T-ALL and predict unfavorable outcomes in pediatric patients [128,129].

### 7.2. Structure and Function

ATM encodes the protein ataxia telangiectasia mutated, which is a serine/threonine protein kinase that plays a key role in DNA damage response [130]. Loss of functional ATM promotes tumorigenesis [131], and mutations in ATM underlie the rare autosomal recessive inherited disorder ataxia-telangiectasia that is characterized by genomic instability, sensitivity to DNA-damaging agents and an increased risk of developing cancer [129]. Although alterations such as those seen here have not been fully characterized and are of unknown functional significance, similar alterations have been previously reported in the context of cancer, which may indicate biological relevance.

### 7.3. Potential Treatment Strategies

Loss of functional ATM results in a defective DNA damage response, and homologous recombination-mediated DNA repair (HRR) predicts sensitivity to PARP inhibitors [132]. Mutations in the HRR genes, including *BRCA1*, *BRCA2* and *ATM*, have been reported in prostate cancer. The PARP enzyme complex is involved in the repair of DNA damage, and its inhibition causes the accumulation of DNA mutations in HRR-deficient cells. Several PARP inhibitors are under development, including talazoparib, niraparib, rucaparib and veliparib [133]. For mCRPC, olaparib was approved by the FDA, with its clinical efficacy demonstrated in a phase III clinical trial [134]. In a phase II study of patients with gastric cancer, the combination of olaparib with paclitaxel resulted in improved overall survival versus paclitaxel alone, both in the overall patient population and the patient population with low ATM protein expression [135].

## 8. JAK2

### 8.1. Frequency and Prognosis

In the AML TCGA dataset, Janus kinase 2 (*JAK2*) mutation was observed in fewer than 1% of samples, and amplification has not been reported [17]. *JAK2* mutation, specifically V617F, is prevalent in classic (BCR-ABL negative) myeloproliferative neoplasms (MPNs) and is reported in up to 95% of polycythemia vera, 50–60% of essential thrombocythemia and 40–50% of primary myelofibrosis cases [136,137,138,139]. Although *JAK2 V617F* mutation did not impact overall survival in AML in a study [140], patients harboring this alteration had a higher relapse rate [141]. Mutations in genes, including *ASXL1*, *CBL*, *DNMT3A*, *MYD88*, *JAK2*, *TET2*, *SF3B1* and *U2AF*1, have been associated with clonal hematopoiesis of indeterminate potential (CHIP), which is age-related and associated with an increased risk of hematologic cancers [142,143,144].

### 8.2. Structure and Function

JAK2 and the additional Janus kinase family members JAK1, JAK3 and TYK2 share a similar structure with seven Janus homology domains [145,146], which constitute the N-terminal FERM and SH2 domain associated with the intracellular domains of cell surface receptors as well as the C-terminal pseudokinase and kinase domain of JAK2. JAK2 encodes Janus kinase 2, a tyrosine kinase that regulates signals triggered by cytokines and growth factors [147]. The *JAK2 V617F* mutation is located on exon 14 in the pseudokinase domain of JAK2 on chromosome 9p24 and consists of a single nucleotide substitution (G1849T), interfering with the basal inhibitory effect of the pseudokinase on the kinase domain, leading to constitutive JAK2 activation. JAK2 also activates the MAPK pathway, which is implicated in many cancers [148] and contains three evolutionarily conserved protein kinases, including RAF, MEK1/2 and ERK1/2, as well as the PI3K/AKT/mTOR pathway [149], which further enhance cell proliferation and survival. JAK2 is often mutated in hematopoietic and lymphoid cancers. The JAK2 alteration observed here has been characterized as activating and is predicted to be oncogenic [136].

### 8.3. Potential Treatment Strategies

On the basis of extensive clinical data on myelofibrosis, a disease type that frequently harbors the *JAK2 V617F* mutation [150,151,152,153], JAK2-activating mutations may predict sensitivity to JAK2 inhibitors such as ruxolitinib. The JAK2 inhibitor fedratinib has been used to treat patients with myelofibrosis, with responses observed for both JAK2 V617F-positive and -negative patients, including patients resistant to ruxolitinib [154,155]. Pacritinib was also recently approved for myelofibrosis in the USA. JAK2-activating alterations may predict sensitivity to HDAC inhibitors [156,157,158]; a phase I/II study of givinostat for patients with JAK2 V617F-mutated polycythemia vera reported ORRs from 72.7% (8/11, 1 CR) to 80.7% (25/31, 3 CRs) across trial arms [159]. Other alterations that activate JAK2, such as fusions [160,161,162,163,164,165] or amplifications [166], may also confer sensitivity to JAK2 inhibitors on the basis of clinical data in myeloid neoplasms as well as preclinical data.

## 9. NTRK3

### 9.1. Frequency and Prognosis

ETS variant gene 6-neurotrophic tyrosine kinase receptor type 3 (ETV6-NTRK3) gene fusion was discovered by a breakpoint analysis of the t(12;15)(p13;q25) translocation associated with congenital fibrosarcoma, a pediatric soft tissue malignancy. ETV6-NTRK3 (EN) encodes the sterile alpha motif oligomerization domain of the ETV6 transcription factor linked to the protein tyrosine kinase domain of the neurotrophin-3 receptor NTRK3. The EN chimeric oncoprotein links to multiple signaling cascades, including RAS/MAPK and PI3K/AKT, through the IRS-1 adapter protein. Recent evidence indicates that a functional insulin-like growth factor 1 receptor axis and higher order polymer formation are essential for EN oncogenesis. EN rearrangements have been reported in case studies of patients with AML [167,168]. The published data investigating the prognostic implications of NTRK3 alterations in AML are limited.

### 9.2. Structure and Function

NTRK3, also known as TRKC, encodes the neurotrophin-3 (NT-3) growth factor receptor, a member of the neurotrophic tyrosine receptor kinase family, including TRKA, TRKB and TRKC; TRKs, which are activated by neurotrophins, play an important role in neuronal survival and differentiation and have also been shown to be involved in oncogenesis in both neurogenic and non-neurogenic cancers [169]. TRK signaling leads to the activation of the RAS/MAPK and PI3K/AKT pathways, and TRKC, as well as the other TRKs, has generally been considered to be an oncogenic receptor; however, studies have found that, in certain contexts, TRKC may function as a tumor suppressor [169,170]. NTRK3 fusions involving a partner gene that is predicted to promote dimerization fused to the kinase domain of NTRK3 [171] have been characterized as activating.

### 9.3. Potential Treatment Strategies

Clinical and preclinical data indicate that NTRK fusions predict sensitivity to TRK inhibitors such as larotrectinib and entrectinib. Larotrectinib is a small-molecule kinase inhibitor that targets NTRK fusions that occur in multiple types of solid tumors. Its FDA approval represents the first instance of a treatment indication being designated as “tumor-agnostic” from the outset, based on actionable genomic insights. An analysis of combined data from several larotrectinib studies reported an ORR of 81% (88/109) in adult and pediatric patients with various solid tumors harboring NTRK fusions treated with larotrectinib; the responses were durable and CRs were observed in 17% of patients [3]. A pooled analysis of three phase I/II trials of entrectinib for adult patients with NTRK fusion-positive solid tumors reported an ORR of 57% (31/54), a median PFS of 11.2 months and a median OS of 20.9 months [172,173]. The preclinical data showed that AML cell lines harboring ETV6-NTRK3 fusion may be sensitive to entrectinib [174]; however, this has not been investigated in the context of other hematological diseases.

## 10. FGFR3

### 10.1. Frequency and Prognosis

The fibroblast growth factor receptor (FGFR) family has emerged as being frequently altered in wide range of tumors, which is likely a result of the involvement of the FGFR pathway in signal transduction pathways regulating cell proliferation, differentiation, and migration, all of which are critical for tumorigenesis [175]. FGFR2 aberrations have been reported at significantly increased rates in cholangiocarcinoma, endometrial carcinoma, and gastric adenocarcinoma, but have not been previously described as a driver in myeloid or lymphoid malignancies. Conversely, FGFR1 abnormalities have been well documented in hematologic malignancies. Hematopoietic neoplasms associated with FGFR1 rearrangement are included in the family of myeloid and lymphoid neoplasms with varying responsiveness to TKIs, though other neoplasms such as AMLs, ALLs, and MPSs with FGFR3 are characterized by aggressive clinical behavior and resistance to imatinib [176]. These malignancies may, however, respond to specific FGFR1/2/3 inhibitors. *FGFR3* mutation has been detected in fewer than 0.5% of AML samples analyzed in COSMIC (Oct 2019). However, published data investigating the prognostic implications of FGFR3 alterations in acute leukemia are limited.

### 10.2. Structure and Function

The fibroblast growth factor (FGF) family comprises 23 highly regulated monomeric proteins that regulate a plethora of developmental and pathophysiological processes, including tissue repair, wound healing, angiogenesis and embryonic development [177]. FGFR is a tyrosine kinase receptor that has five distinct receptors (FGFR1-5) encoded by the FGFR1-4 and FGFRL1 genes. The binding of FGF to FGFR is facilitated by the glycosaminoglycan heparin. Activated FGFRs phosphorylate tyrosine kinase residues that mediate the induction of downstream signaling pathways, such as RAS/MAPK, PI3K/AKT, PLCγ and STAT [178,179,180]. Dysregulation of FGF/FGFR signaling occurs frequently in cancer due to gene amplification, FGF-activating mutations, chromosomal rearrangements, integration and oncogenic fusions. Aberrant FGFR signaling also affects organogenesis, embryonic development and tissue homeostasis, and it has been associated with cell proliferation, angiogenesis, cancer and other pathophysiological changes. FGFR3 alterations that disrupt the C-terminal region after the kinase domain have been demonstrated to be transforming, with a significantly higher efficiency than full-length FGFR3 [181,182], although they did not induce high levels of constitutive FGFR3 activation [183].

### 10.3. Potential Treatment Strategies

Alterations that activate FGFR3 may predict sensitivity to FGFR inhibitors, including pan-FGFR inhibitors such as erdafitinib [184,185] and infigratinib [186]; multikinase inhibitors pazopanib [187,188] and ponatinib [189]; or antibodies targeting FGFR3 such as vofatamab [190]. Pan-FGFR inhibitors, such as rogaratinib and infigratinib, have predominantly been studied and have shown activity in the context of urothelial carcinoma, where ORRs of ~25% and DCRs of 64–73% have been reported in patients with FGFR mRNA overexpression or FGFR3 alteration [186]. For infigratinib, responses were not observed in patients with non-urothelial tumor types harboring *FGFR3* mutation or amplification [191,192]. The promising results of pemigatinib and infigratinib in advanced unresectable cholangio-carcinoma harboring FGFR2 fusions or rearrangement, and erdafitinib in metastatic urothelial carcinoma with FGFR2 and FGFR3 genetic aberrations, led to their accelerated approval by the FDA. Along with these agents, many phase II/III clinical trials are currently evaluating the use of derazantinib, infigratinib and futibatinib either alone or in combination with immunotherapy. As for hematological malignancies, the FGFR1/2/3 inhibitor pemigatinib has shown impressive efficacy in a small registration-directed trial in patients with FGFR1-rearranged myeloid/lymphoid neoplasms [193].

## 11. EGFR

### 11.1. Frequency and Prognosis

Epidermal growth factor receptor (EGFR) mutation was detected in <1% of AML samples analyzed in COSMIC (Aug 2020) [194]. The published data investigating the prognostic implications of EGFR mutation in AML are limited, though some studies suggest that EGFR expression is intimately associated with poor clinical outcomes [195]. One study confirmed the presence of EGFR in AML and indicated that EGFR expression confers poor prognosis in AML. Overall survival was significantly shorter in 21 patients with EGFR, with an average survival of 18.57 months compared with 31.27 months in 39 patients without EGFR. However, the underlying cause of this adverse prognostic effect has not been identified [196].

### 11.2. Structure and Function

EGFR belongs to a class of proteins called receptor tyrosine kinases. In response to signals from the environment, EGFR passes biochemical messages to the cell that stimulate it to grow and divide [197]. The expression of EGFR, an important proto-oncogene, regulates cell differentiation, proliferation, cell migration and survival in most cancer types. EGFR expression also has been reported in AML. EGFR is a cell surface receptor belonging the ErbB family of proteins. The binding of ligands (EGF) activates EGFR though dimerization, leading to the activation of other downstream pathways such as RAS/MAPK and PI3K/AKT through signal transduction [198]. Mutations of EGFR lead to constitutive receptor activation. Overexpressed EGFR results in aberrant signaling that induces uncontrolled cell growth and oncogenesis [199]. The expression of EGFR is not uniform across all cell types. Overexpression or mutation of the *EGFR*/*ERBB* gene has been detected in a variety of human solid tumor cells, such as epithelial, mesenchymal, neuroectodermal and AML cells [200].

### 11.3. Potential Treatment Strategies

EGFR-activating mutations may predict sensitivity to EGFR TKIs, including erlotinib [197], gefitinib [201], afatinib [202], dacomitinib [203] and osimertinib [204]. Third-generation EGFR inhibitors, such as osimertinib, selectively target mutated EGFR, including EGFR T790M [204,205]. Two recent clinical case reports showed that erlotinib caused the complete remission of AML-M1 in patients with both AML-M1 and non-small-cell lung cancer [206]. These results are supported by preclinical studies, in which EGFR TKIs have anti-proliferative effects on AML. Recent studies have emphasized that EGFR inhibitors in combination with other potential drugs may prove to be useful against hematological malignancy [200].

## 12. Conclusions and Perspectives

Various factors and mechanisms have been revealed to influence cancer development, including gene signaling, genome instability, cancer immunity, growth suppression systems and angiogenesis. Moreover, a variety of drugs have been developed for solid tumors, for example, TKIs, PARP inhibitors, ICIs, cyclin-dependent kinase (CDK) inhibitors and vascular endothelial growth factor (VEGF) inhibitors. We listed the combinations of detectable genes and FDA-approved drugs in Table 1; however, many more molecular-targeted therapies exist. As mentioned in the Introduction, the fact that hematological tumors have mechanisms that are distinct from solid tumors may be one of the reasons that effective tumor-agnostic therapies are not more advanced. It remains possible that other undiscovered mechanisms may exist. However, CGPs detect various kinds of cancer-related gene alterations, including short variants, fusions, amplifications, rearrangements, overexpression and others. The accumulated information about these gene alterations is anticipated to contribute to clarifying the mechanisms of AML, leading to the rapid development of tumor-agnostic therapies in the near future.

## Figures and Tables

**Figure 1 biomedicines-10-03008-f001:**
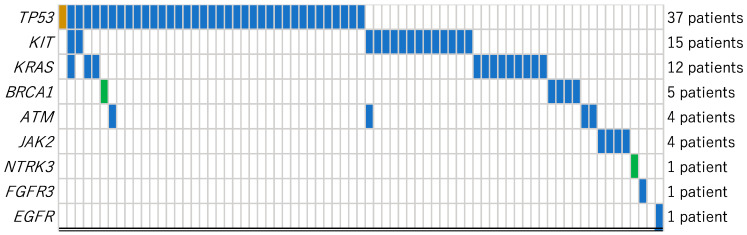
Detailed analysis of the published data of (HM)-SCREEN-Japan 01 [18,19]: Frequencies of tumor-agnostic variants detected among 177 patients with relapsed/refractory or newly diagnosed unfit AML. *TP53* (thirty-seven cases), *KIT* (fifteen cases), *KRAS* (twelve cases), *BRCA1* (five cases), *ATM* (four cases), *JAK2* (four cases), *NTRK3* (one case), *FGFR3* (one case) and *EGFR* (one case) are typical genes for which molecular-targeted drugs have been developed or are under development for solid tumors. They are mostly exclusive of each other. Frequent variants are *TP53*, *KIT* and *KRAS*. TP53 accounts for half of these mutations and 21% of all the examined samples.

**Figure 2 biomedicines-10-03008-f002:**
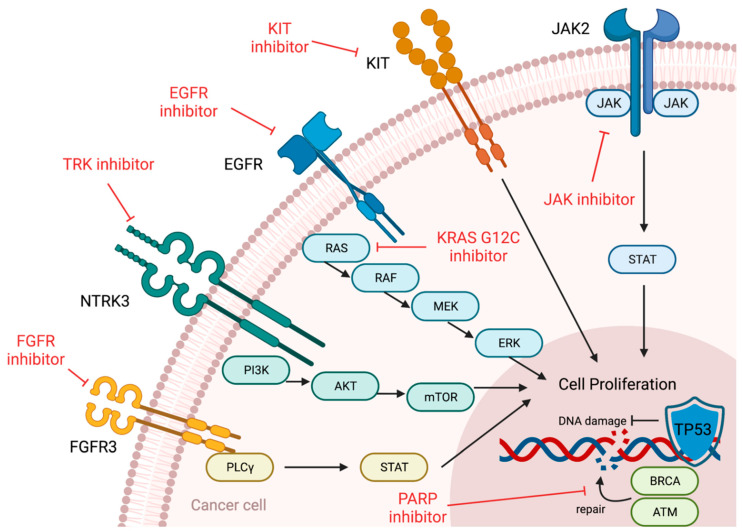
Schematic drawing of cell proliferation. JAK2, KIT, EGFR, NTRK3 and FGFR3 proteins are tyrosine kinase receptors. KRAS proteins transduce receptor signaling into cellular processes. These proteins activate the signaling pathways such as PI3K/AKT/mTOR and RAS/RAF/MEK/ERK pathways. BRCA and ATM have an important role in repairing damaged DNA. Molecular-targeted drugs suppress these proteins overexpressed by gene mutations. TP53 maintains the harmony between cell arrest and growth during genomic stress.

**Figure 3 biomedicines-10-03008-f003:**
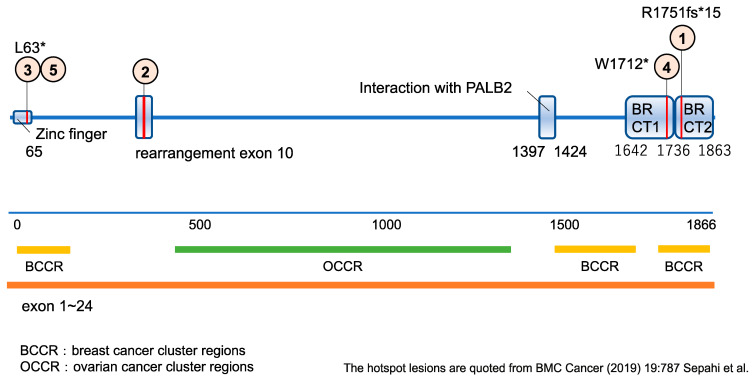
The diagram of a BRCA1 protein structure [120]: It is composed of 24 exons and 1866 Amino acids. In (HM)-SCREEN-Japan, five cases with pathogenic or germline BRCA1 variants were detected. Positions of these variants are shown above with red vertical lines on the BRCA1 protein. There are a number of hotspots where cancer related genes are located. BCCR and OCCR are the hotspots where a majority of hereditary breast and ovarian cancer related genes are located. * Case 1, 3, 4 and 5 were located on the BCCRs and case 2 was considered as pathogenic according to ClinVar, which suggested the BRCA gene variants might be related to AML development.

**Table 1 biomedicines-10-03008-t001:** Tumor-agnostic mutations and molecular-targeted drugs approved by the Food and Drug Administration (FDA).

Biomarkers	Tumor Types	Common Alterations	Drugs
*EGFR*	NSCLC	exon 19 deletions, exon 21 L858R, exon 20 insertions and T790M	Gefitinib, erlotinib, osimertinib, dacomitinib, afatinib, mobocertinib and amivantamab
*ALK*	NSCLC	ALK protein expression, *ALK* gene rearrangements	Alectinib, brigatinib, lorlatinib, ceritinib and crizotinib
*ROS1*	NSCLC	*ROS1* fusions	Crizotinib and entrectinib
*RET*	NSCLC	*RET* fusions	Pralsetinib and selpercatinib
	Thyroid	*RET* fusions, mutations	Selpercatinib
*MET*	NSCLC	*MET* exon 14 skipping	Capmatinib
*RAS*	NSCLC	*KRAS* G12C	Sotorasib
	Colorectal	KRAS wild-type, NRAS wild-type	Cetuximab and panitumumab
*BRAF*	NSCLC, Thyroid	V600E	Dabrafenib and trametinib
	Colorectal	V600E	Encorafenib
	Melanoma	V600E, V600K	Encorafenib, binimetinib, cobimetinib, vemurafenib, trametinib and dabrafenib
*ERBB2*	Breast	HER-2 protein overexpression, *ERBB2* gene amplification	Trastuzumab, trastuzumab emtansine, trastuzumab deruxtecan, pertuzumab, lapatinib, neratinib, tucatinib and margetuximab
	Gastric	HER-2 protein overexpression, *ERBB2* gene amplification	Trastuzumab, trastuzumab deruxtecan and margetuximab
*KIT*	GIST and Mastocytosis	D816V, Kit protein expression	Imatinib
*PIK3CA*	Breast Cancer	C420R, E542K, E545A, E545D, E545G, E545K, Q546E, Q546R, H1047L, H1047R and H1047Y	Alpelisib
*FGFR2*	Cholangiocarcinoma	FGFR2 fusions, rearrangements	Pemigatinib and infigratinib
*FGFR3*	Urothelial Cancer	Exon 7: R248C, S249C; exon 10: G370C, Y373C and fusions	Erdafitinib
NTRK1/2/3	Any Solid Tumors	NTRK1, NTRK2 and NTRK3 fusions	Larotrectinib and entrectinib
*JAK2*	MPNs	V617F	Ruxolitinib, fedratinib and pacritinib
*EZH2*	Follicular Lymphoma	Y646N, Y646F, Y646X, Y646H, Y646S, Y646C, A682G and A692V	Tazemetostat
*BCR-ABL1*	CML	Major *BCR-ABL* fusion	Imatinib, nilotinib, dasatinib, bosutinib, ponatinib and asciminib
	Ph + ALL	Minor *BCR-ABL* fusion	Imatinib, dasatinib and ponatinib
*BRCA1/2*	Breast, Ovarian, Pancreatic Cancer and mCRPC	Deleterious/suspected deleterious mutations in *BRCA1*/*2* genes	Olaparib, talazoparib (breast), rucaparib (mCRPC, ovarian) and niraparib (ovarian)
HRR	mCRPC	*BRCA1*, *BRCA2*, *ATM*, *BARD1*, *BRIP1*, *CDK12*, *CHEK1*, *CHEK2*, *FANCL*, *PALB2*, *RAD51B*, *RAD51C*, *RAD51D* and *RAD54L* alterations	Olaparib
MMR	Any Solid Tumors	MLH1, PMS2, MSH2 and MSH6 mutation	Dostarlimab-gxly
TMB	Any Solid Tumors	TMB ≥ 10 mutations per megabase	Immune checkpoint inhibitors
MSI		Microsatellite instability–high	

Abbreviations: *ALK* (anaplastic lymphoma kinase gene)*, ROS1* (ros proto-oncogene 1), *RET* (ret proto-oncogene), *MET* (met proto-oncogene), *RAS* (rat sarcoma viral oncogene homologue gene), *BRAF* (b-raf proto-oncogene), *ERBB2* (erb-b2 receptor tyrosine kinase 2 gene), *PIK3CA* (phosphatidylinositol-4,5-bisphosphate 3-kinase catalytic subunit alpha), *EZH2* (enhancer of zeste 2 polycomb repressive complex 2 subunit), *BCR-ABL1* (BCR activator of RhoGEF and GTPase- ABL proto-oncogene 1, non-receptor tyrosine kinase), HRR (homologous recombination repair), MMR (mismatch repair), TMB (tumor mutation burden), MSI (Microsatellite Instability), NSCLC: non-small-cell lung cancer, GIST: gastrointestinal stromal tumor, mCRPC: metastatic castrate resistant prostate cancer, MPNs: myeloproliferative neoplasms, CML: chronic myeloid leukemia, Ph + ALL: Philadelphia chromosome-positive acute lymphoblastic leukemia.

## Data Availability

Data sharing not applicable.

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
