# Peer review of "Molecular-Targeted Therapy for Tumor-Agnostic Mutations in Acute Myeloid Leukemia"

_biomedicines, 2022, doi:10.3390/biomedicines10123008_

Round 1

Reviewer 1 Report (Previous Reviewer 1)

Please provide point by point responses to my comments bellow. (Your response after every concern raised).

Furthermore, please highlight in the manuscript what is corrected in comparison to previous version according to the reviewers' comments.

1. NTRK gene fusion in introduction – name the full name of the gene when first mentioned

2. Larotrecinib and entrectinib in introduction – when first mentioning write that their relation to NTTRK is (inhibition…?)

3. Full gene names should be stated in the text and/or table

4. What about biallelic (multi-hit) TP53 mutations – acute erythroid leukemia has high prevalence of biallelic TP53 mutation. What about therapy related AML and TP53 as well as worse prognosis associated with biallelic mutation? This is important in context of therapy since MDS-biTP53 might be therapeutically treated as AML (PMID: 32747829). Please also see the new WHO classification of myeloid neoplasms (PMID: 35732831)

5. Again „there is no data supporting the use of pazopanib and regorafenib for KITactivated hematologic tumors…” – what are pazopanib and regorafenib – please mention what they do regarding KIT. Same with sorafenib etc throughout the manuscript…

6. Explain why are these nine genes identified? Why there is no mention of IDH gene mutation – as I can see in the work by Hosono et al, the IDH mutation in de novo AML is the ninth according to frequency in the population studied? Please if these studies by Hosono et al. and Miyamoto et al. are congress abstracts, and you are basing your current work on them, highlight that information. Why did you include BRCA1 in your review if there is such a low frequency in AML?

7. The authors should focus more on the AML mutations since that is the title of their manuscript – there are lots of examples of mutations in solid tumours. Furthermore, there is not only acute myeloid leukemia in the manuscript but also other hematological malignancies – the title does not follow the manuscript content.

8. Figure 2 – what is the reference of this figure? Is this from the original article that you should have cited here? There are no references in the text concerning this figure. It is a bit odd that the authors have included their findings in the review article form – if you find it important, please explain your rationale for doing so.

9. It would be helpful for the readership if gene targets and downstream signaling of the important genes in AML could be summed up in the new figure.

10. Are there examples of different genetic AML (or solid tumor) mutations that could be treated with the combination of the drugs, eg. FLT3 and mutated IDH inhibitors?

Author Response

Reviewer 1

  1. NTRK gene fusion in introduction – name the full name of the gene when first mentioned

Thank you for your kind advice. We added the explanation that NTRK is the abbreviation of neurotrophic tyrosine receptor kinase 3 gene in the revised manuscript.

  1. Larotrecinib and entrectinib in introduction – when first mentioning write that their relation to NTTRK is (inhibition…?)

The development of a therapeutic agent effective across solid tumors, not only a single cancer type, is a breakthrough, and we wanted to show that these targeted genes had the potential to eventually be approved in hematologic tumors, including AML. Larotrecinib and entrectinib, as inhibitors to NTRK fusion, represent the dawn of a new era of tumor-agnostic molecular targeted therapies.

  1. Full gene names should be stated in the text and/or table

Thank you, I revised them.

  1. What about biallelic (multi-hit) TP53 mutations – acute erythroid leukemia has high prevalence of biallelic TP53 mutation. What about therapy related AML and TP53 as well as worse prognosis associated with biallelic mutation? This is important in context of therapy since MDS-biTP53 might be therapeutically treated as AML (PMID: 32747829). Please also see the new WHO classification of myeloid neoplasms (PMID: 35732831)

We are so glad you kindly gave some references. We added the sentences below in TP53 section, according to your thoughtful advice.

Overall survival (OS) and AML transformation in MDS were significantly different between the TP53 allelic state. In multi-hit state, the median OS was 8.7 months, compared with 2.5 years in monoallelic patients. TP53 mutations are more common in therapy-related MDS (t-MDS) and are associated with a high risk of progression to AML. TP53-mutated t-MDS patients more frequently had multiple hits compared to TP53-mutated de novo patients (84 versus 65%). Patients with acute erythroid leukemia (AEL) frequently had TP53 gene mutations (25-36%) and biallelic mutations are a feature of a subset of AEL . TP53-mutations were associated with very poor prognosis as well in AEL (median survival of 13 months, with no patients surviving at 5 years)

  1. Again „there is no data supporting the use of pazopanib and regorafenib for KITactivated hematologic tumors…” – what are pazopanib and regorafenib – please mention what they do regarding KIT. Same with sorafenib etc throughout the manuscript…

Pazopanib and regorafenib are multi-kinase inhibitors. KIT is included as one of their targeted proteins. Unfortunately, they are not proven to have clinical benefit even in studies for solid tumors. But recently, molecular targeted drugs have been developed for most of cancer-related genes. So, I hope drugs targeting KIT can be approved for hematological malignancies in the near future. However, it is confusing to mention the drugs with no clinical evidence, so I deleted the sentences about pazopanib and regorafenib. However, I insist it is important to mention sorafenib because preclinical and limited clinical data indicated that KIT mutations were sensitive to sorafenib. The purpose of this review is to highlight the tumor-agnostic mutations for which many drugs have clinical benefit in solid tumors and to consider the possibility the drugs targeting such mutations detected in AML can be beneficial for hematological malignancies as well.

  1. Explain why are these nine genes identified? Why there is no mention of IDH gene mutation – as I can see in the work by Hosono et al, the IDH mutation in de novo AML is the ninth according to frequency in the population studied? Please if these studies by Hosono et al. and Miyamoto et al. are congress abstracts, and you are basing your current work on them, highlight that information. Why did you include BRCA1 in your review if there is such a low frequency in AML?

We intentionally excluded the mutations for which some drugs were recognized effective in AML. We didn’t intend to summarize molecular targeted drugs which were already recognized beneficial for AML patients. We wanted to focus on tumor-agnostic mutations which were detected among solid tumors and AML referring to (HM)-SCREEN Japan01, because molecular targeted therapies for solid tumors can be effective for AML as well. We are sure considering tumor-agnostic mutations will help to develop new drugs for AML patients.

Although BRCA1 is not frequently detected in AML, it is important to consider its significance. Because the management of hereditary cancers has attracted wide attention in solid tumors. If germline or pathogenic BRCA1 develop AML, PARP inhibitors can be strong candidates for AML treatment. At the same time, we have to consider the management of hereditary AML. therefore, we considered the significance of BRCA1.

  1. The authors should focus more on the AML mutations since that is the title of their manuscript – there are lots of examples of mutations in solid tumours. Furthermore, there is not only acute myeloid leukemia in the manuscript but also other hematological malignancies – the title does not follow the manuscript content.

Tumor-agnostic mutation means the mutations for which specific molecular targeted drugs are effective in a variety of cancer types across organs. We didn’t intend to discuss the mutations already clinically targetable in AML. But in the first manuscript, we mentioned IDH1/2 and FLT3 inconsistently, so I deleted the sentences.

  1. Figure 2 – what is the reference of this figure? Is this from the original article that you should have cited here? There are no references in the text concerning this figure. It is a bit odd that the authors have included their findings in the review article form – if you find it important, please explain your rationale for doing so.

Figure 2 was referred to (HM)-SCREEN-Japan01, which is an actionable mutation profiling multicenter study of patients with newly diagnosed AML who cannot be treated with first standard treatment or patients who have relapsed/refractory AML (R/R-AML)

  1. Hosono, N., et.al., Hematologic Malignancies (HM)-Screen-Japan 01: A Mutation Profiling Multicenter Study on Patients with Acute Myeloid Leukemia. Blood, 2021(138): p. 4457.
  2. Miyamoto, K., et. al., Interim Analysis of Hematologic Malignancies (HM)-Screen-Japan 01: A Mutation Profiling Multicenter Study of Patients with AML. Blood, 2020. 136: p. 2–3.

The objective of this study was to evaluate the frequency and characteristics of cancer-related genomic alterations in patients with AML using a comprehensive genome profiling assay. We explained in the revised manuscript that we had adopted the nine tumor-agnostic mutations referring to this study. We thought Figure 2 would help to consider if pathogenic and germline BRCA1 mutations could develop AML.

  1. It would be helpful for the readership if gene targets and downstream signaling of the important genes in AML could be summed up in the new figure.

Thank you for your great advice. We added new figure showing the functions of the nine genes. We also showed how the molecular targeted drugs regulate tumor development.

  1. Are there examples of different genetic AML (or solid tumor) mutations that could be treated with the combination of the drugs, eg. FLT3 and mutated IDH inhibitors?

The representative mutations in AML such as IDH and FLT3 are not our subject. We wish your understanding.

Reviewer 2 Report (New Reviewer)

Manuscript ID: biomedicines-2033464

Title: Molecular Targeted Therapy for Tumor-agnostic Mutations in Acute Myeloid Leukemia

In this review article, the authors summarized the role of mutations on cancer-related genes, including TP53, KIT, KRAS, BRCA1, ATM, JAK2, NTRK3, FGFR3 and EGFR genes in Acute Myeloid Leukemia (AML). The authors select those genes according to the comprehensive genomic profiling (CPG) study conducted in Japan. At this time (01 November, 2022), a search in PubMed, using the key words “acute myeloid leukemia, mutations” includes 19,864 results, while the key words “acute myeloid leukemia, mutations, review” 3,396 results, respectively. I believe that this review article does not add original information regarding the topic concerning the role of mutations on cancer-related genes in AML comparison with other authoritative reviews published on this topic.

Minor comments:

(1).   Introduction: It is highly recommended to use a sentence to specify the aim of this review article.

(2).   The authors listed the combinations of detectable genes and FDA approved drugs in Table 1. How to access approved drug from FDA databases?

(3).    6. BRCA1: I think the title should be BRCA1 and BRCA2.

Author Response

Reviewer 2:

In this review article, the authors summarized the role of mutations on cancer-related genes, including TP53, KIT, KRAS, BRCA1, ATM, JAK2, NTRK3, FGFR3 and EGFR genes in Acute Myeloid Leukemia (AML). The authors select those genes according to the comprehensive genomic profiling (CPG) study conducted in Japan. At this time (01 November, 2022), a search in PubMed, using the key words “acute myeloid leukemia, mutations” includes 19,864 results, while the key words “acute myeloid leukemia, mutations, review” 3,396 results, respectively. I believe that this review article does not add original information regarding the topic concerning the role of mutations on cancer-related genes in AML comparison with other authoritative reviews published on this topic.

Thank you for your good suggestion. We discussed the tumor-agnostic mutations in AML. The development of CGP revealed that the representative mutations in solid tumors were also detected in AML, which suggested molecular targeted therapies for solid tumors could be effective for AML. At present, there are few molecular targeted therapies for AML unlike for solid tumors. We hope a variety of new strategies for AML will appear targeting a large amount of well-known mutations of solid tumors.

Minor comments:

(1).   Introduction: It is highly recommended to use a sentence to specify the aim of this review article.

Thank you for your kind advice, in the Introduction part, we added the sentence as below;

In this review, we summarized the frequency, the prognosis, the structure and the function of these mutations as well as current treatment strategies in solid tumors, and revealed genetical relations between solid tumors and AML and to develop tumor agnostic molecular targeted therapies and lifetime management strategies in AML.

(2).   The authors listed the combinations of detectable genes and FDA approved drugs in Table 1. How to access approved drug from FDA databases?

I referred to an official website of the United States government: Drug Approvals and Databases

https://www.fda.gov/drugs/development-approval-process-drugs/drug-approvals-and-databases

(3).    6. BRCA1: I think the title should be BRCA1 and BRCA2.

In (HM)-SCREEN-Japan01, we detected BRCA1 but didn’t detect BRCA2. Although BRCA1 and 2 have similar mechanism, we should focus on the detected mutations. However, we mentioned BRCA2 in some sentences inappropriately, so we deleted them according to your advice.

Round 2

Reviewer 1 Report (Previous Reviewer 1)

I have no further suggestions.

Reviewer 2 Report (New Reviewer)

The authors responded to each question from reviewers improving the content of text. I recommend this paper to be accepted.

This manuscript is a resubmission of an earlier submission. The following is a list of the peer review reports and author responses from that submission.

Round 1

Reviewer 1 Report

In the manuscript „Potentially Targetable Tumor-agnostic Variants in Acute Myeloid Leukemia“ authors review the current knowledge of some of gene mutations that came up in the CGP study of patients with relapsed/refractory or newly diagnosed unfit AML conducted across Japanese institutes.

Broad comments: The manuscript is readable and reports on the important matter, but the execution is not the best so there a few things that have to be improved. 

Specific comments:

1.     NTRK gene fusion in introduction – name the full name of the gene when first mentioned

2.     Larotrecinib and entrectinib in introduction – when first mentioning write that their relation to NTTRK is (inhibition…?)

3.     Full gene names should be stated in the text and/or table

4.     What about biallelic (multi-hit) TP53 mutations – acute erythroid leukemia has high prevalence of biallelic TP53 mutation. What about therapy related AML and TP53 as well as worse prognosis associated with biallelic mutation? This is important in context of therapy since MDS-biTP53 might be therapeutically treated as AML (PMID: 32747829). Please also see the new WHO classification of myeloid neoplasms (PMID: 35732831)

5.     Again „there is no data supporting the use of pazopanib and regorafenib for KIT-activated hematologic tumors…” – what are pazopanib and regorafenib – please mention what they do regarding KIT. Same with sorafenib etc throughout the manuscript…

6.     Explain why are these nine genes identified? Why there is no mention of IDH gene mutation – as I can see in the work by Hosono et al, the IDH mutation in de novo AML is the ninth according to frequency in the population studied?

Please if these studies by Hosono et al. and Miyamoto et al. are congress abstracts, and you are basing your current work on them, highlight that information.  

Why did you include BRCA1 in your review if there is such a low frequency in AML?

7.     The authors should focus more on the AML mutations since that is the title of their manuscript – there are lots of examples of mutations in solid tumours. Furthermore, there is not only acute myeloid leukemia in the manuscript but also other hematological malignancies – the title does not follow the manuscript content. 

8.     Figure 2 – what is the reference of this figure? Is this from the original article that you should have cited here? There are no references in the text concerning this figure. It is a bit odd that the authors have included their findings in the review article form – if you find it important, please explain your rationale for doing so.

9.     It would be helpful for the readership if gene targets and downstream signaling of the important genes in AML could be summed up in the new figure. 

10.  Are there examples of different genetic AML (or solid tumor) mutations that could be treated with the combination of the drugs, eg. FLT3 and mutated IDH inhibitors? 

Reviewer 2 Report

It is confusion to know what the point of this manuscript is.  The authors mention comprehensive genomic profiling (CGP) relative to tumor agnostic therapeutics (the first 2 approvals in cancer across histologic diagnoses being ICIs and NTRK axis agents) and lament the fact that there are not tumor agnostic targets in AML.  They then show a table irrelevant to that discussion mainly containing diagnosis specific inhibitors in cancer and do not emphasize the fact that IDH1/IDH2 inhibitors are approved based on molecular profiles in AML and may be used in other malignancies.  The authors then show a panel of mutations from a previous Japanese study but do not justify how these where selected or whether there are tiers of actionability, etc as in published guidelines elsewhere.  Finally, there is only potential inhibitors mentioned rather than patients treated successfully with such a targeted approach.  For the most common alteration here, TP53 there basically is no potential treatment(s) mentioned, thereby defeating the purpose of most of the manuscript.

Whether or not germline alterations were uncovered in this study would have also been interesting and lacking here.

For all of these reasons I do not think any of the presented manuscript advances anything in the field.